# Impact of Rainfall on the Detection Performance of Non-Contact Safety Sensors for UAVs/UGVs

**DOI:** 10.3390/s24092713

**Published:** 2024-04-24

**Authors:** Yasushi Sumi, Bong Keun Kim, Takuya Ogure, Masato Kodama, Naoki Sakai, Masami Kobayashi

**Affiliations:** 1National Institute of Advanced Industrial Science and Technology (AIST), Tsukuba 305-8568, Japan; 2Altech Corporation, Yokohama 220-6218, Japan; 3National Research Institute for Earth Science and Disaster Resilience (NIED), Tsukuba 305-0006, Japan

**Keywords:** safety sensor, performance evaluation, object detection, UAV, UGV, rainfall, spatial transmittance, drop size distribution, precipitation rate

## Abstract

This study comprehensively investigates how rain and drizzle affect the object-detection performance of non-contact safety sensors, which are essential for the operation of unmanned aerial vehicles and ground vehicles in adverse weather conditions. In contrast to conventional sensor-performance evaluation based on the amount of precipitation, this paper proposes spatial transmittance and particle density as more appropriate metrics for rain environments. Through detailed experiments conducted under a variety of precipitation conditions, it is shown that sensor performance is significantly affected by the density of small raindrops rather than the total amount of precipitation. This finding challenges traditional sensor-evaluation metrics in rainfall environments and suggests a paradigm shift toward the use of spatial transmittance as a universal metric for evaluating sensor performance in rain, drizzle, and potentially other adverse weather scenarios.

## 1. Introduction

The aim of this paper is to explore a method for quantitatively assessing how rain and drizzle affect the detection capabilities of non-contact safety sensors in adverse weather conditions. In recent years, unmanned aerial vehicles (UAVs) and unmanned ground vehicles (UGVs), such as drones, delivery robots, autonomous vehicles, and agricultural machines, have reached the stage of practical application. These autonomous systems, which coexist with humans, must incorporate non-contact safety sensors to accurately perceive their surroundings, detect approaching people or hazardous objects, and perform safe stopping or avoiding maneuvers.

Outdoor operations present unique challenges, such as light interference from the sun and reduced visibility during adverse weather (rain, snow, fog, etc.), which can affect sensors’ detection capabilities, as illustrated in Figure 1. Consequently, sensors that maintain performance integrity in outdoor environments are essential. However, robust sensor technology alone is not enough. Establishing precise evaluation criteria and test methodologies is crucial to ensure consistent performance under all expected environmental conditions.

This study focuses on the effects of rain and drizzle, the most common severe weather events in warm, humid climates. For safety sensors designed to operate in rainy conditions, it is important to inform users of the operational limitations associated with rainfall. Specifically, it must be clearly specified under which conditions the sensor retains the ability to detect objects and measure distance and when these functions become compromised.

The precipitation rate is a metric that has been used to evaluate the performance of sensors in rainfall environments. However, this paper argues that precipitation rates may not accurately reflect sensor detection capabilities. For example, a sensor may work properly in heavy rain (80 mm/h) but fail in light drizzle (2 mm/h).

The main contributions of this paper include demonstrating that evaluating sensors based solely on precipitation rate can be misleading in certain scenarios and proposing a more effective method for performance evaluation based on spatial transmittance, or raindrop density. Using this novel metric, we quantitatively demonstrate the varying impacts of different rainfall types on sensor efficiency and explore the behavior of common sensing technologies in rainfall conditions.

This paper is organized as follows: After clarifying the scope of this paper in Section 1.1, we first outline the physical properties of rainfall based on meteorological studies in Section 2.1. We then review relevant international standards and core sensing technologies in rainfall environments and present our previous work on sensor evaluation techniques in adverse weather conditions in Section 2.2, Section 2.3 and Section 2.4.

In Section 3, we propose a method for evaluating sensor performance in rain and drizzle environments. First, in Section 3.1, we compare precipitation rates and spatial transmittance as evaluation metrics. The problems associated with evaluating sensors based on precipitation rate are described, and the superiority of spatial transmittance is demonstrated. Section 3.2 then defines the object-detection rate in a precipitation environment, and Section 3.3, Section 3.4 and Section 3.5 detail the predefined precipitation conditions reproduced in the experiment, the test piece and experimental setup, and the sensor device under test.

Section 4 presents the experimental results, including measurements of the physical properties of the precipitation conditions and the object-detection rates of the sensors under these conditions. It also discusses the impact of precipitation on detection performance. Finally, we discuss metrics for evaluating sensor performance in rain and drizzle environments in Section 5. Section 6 is the conclusion.

Our experimental results lead us to conclude that while small raindrops primarily affect visibility, large raindrops affect precipitation rates. Therefore, as a metric for evaluating sensor performance, we recommend using the spatial transmittance or the particle density of the rain environment.

### 1.1. Scope

Before considering the safety of machines, including UAVs and UGVs, it is important to clearly define the scope of the application [1]. In this paper, we will focus on the behavior of sensors in rain and drizzle as meteorological phenomena. We will not discuss fog or mist, which are caused by the same water particles.

Water droplets on the optical windows of sensors will not be discussed in this paper. Although water droplets on windows have a significant impact on sensor performance, this issue relates to the “pollution” of the optical window and should be distinguished from the effects of raindrops or drizzle in the observation space. The issue of the optical window pollution is covered in detail in the IEC 61496 series, which specifies detailed design requirements and test methods [2].

The effect of wind on rain and drizzle, while potentially significant, is not treated here and is left for future study.

This paper focuses on the evaluation of non-contact safety sensors based on optical principles, similar to those discussed in our previous paper [3]. Specifically, we examine “depth cameras” (also known as 3D cameras, distance imaging cameras, or RGBD sensors). These cameras capture scenes at relatively short distances as depth images or 2D arrays, where each pixel contains distance information from the point of measurement. They are relatively inexpensive and can also capture RGB images with the same field of view as depth images, making them highly compatible with deep-learning-based object recognition and identification. As such, they are expected to find applications in both safety and non-safety systems across a variety of fields.

This paper covers the three main sensing principles used in depth camera products: Time of Flight, Stereo Vision, and Structured Light. Time-of-Flight (TOF) sensors calculate the distance to an object by measuring the time it takes for emitted periodic light to reflect back from the object. Both Stereo Vision and Structured Light sensors use triangulation, a principle of multiple-view geometry [4], to determine the distance to feature points on an object. We distinguish between systems that use two or more cameras for triangulation, referred to as Stereo Vision (SV), and those that combine a single camera with a pattern light projector, referred to as Structured Light (SL) [5]. Some SV-based sensors incorporate an auxiliary pattern light projector for enhanced performance.

This paper does not specifically address LiDAR technology, which measures distance over medium to long ranges by planar or spatial scanning using single or multiple laser beams. However, since LiDAR- and TOF-based depth cameras share similar sensing principles, the discussion in this paper is likely to be applicable to LiDAR in many scenarios. Radar and ultrasonic sensors are outside the scope of this paper due to their use of different sensing wavelengths, distance ranges, and the relatively low resolution of their measurements. Similarly, proximity and contact sensors are not discussed as they do not align with the paper’s focus on optical sensing technologies.

## 2. Background and Related Work

### 2.1. Meteorology

#### 2.1.1. Precipitation

Precipitation as a meteorological observation value, also known as precipitation rate, intensity, or amount, is defined as “the volume of water, usually measured in millimeters or inches, that falls to the ground at a given location during a given period of time” [6]. This metric, widely accepted and used in daily weather forecasting, offers a straightforward measure of rainfall amount and intensity. Consequently, it has been adopted as a metric to evaluate the performance of outdoor safety sensors, typically to denote the operational limits of the sensor.

However, this paper argues against the use of precipitation rate as a metric for sensor detection capability. The rationale is that precipitation rate does not necessarily correlate accurately with sensor performance. In some cases, it can mislead by misrepresenting sensor performance. Sensor performance should be evaluated by the physical characteristics of the precipitation space from the sensor to the object, not by the rate of precipitation accumulated on the ground.

#### 2.1.2. Particle Density and Drop Size Distribution

Precipitation, a meteorological phenomenon in which liquid water droplets fall from clouds to the ground, is conveniently divided into drizzle and rain based on particle size. Drizzle is precipitation consisting of water particles 0.1 to 0.5 mm in diameter [7]. Particles smaller than 0.1 mm in diameter do not reach the ground because they evaporate during their descent from the cloud. Rain, on the other hand, is precipitation composed mainly of water particles larger than 0.5 mm in diameter, with the maximum particles being 5 to 8 mm in diameter. Water droplets larger than this are not stable and break up as they fall and do not occur naturally [8]. Note that normal rainfall also includes drizzle-sized particles, which are more numerous than large raindrops.

The particle sizes, along with their fall velocities, are measured by a disdrometer, a meteorological instrument. This instrument records the particle size and fall velocity of raindrops as they pass through its measurement area, which is defined by a line laser beam plane, and outputs these data as a particle size and velocity distribution (PSVD) matrix [9]. This matrix is a 2D frequency distribution that details the discrete information of the particle size and drop velocity distribution. It is presented as a 2D matrix that records the cumulative number of raindrops per unit time for raindrops of a given particle size class $D_{i}$ and a velocity class $V_{j}$. Figure 2a shows a heat map visualizing the PSVD matrix from a 300 mm/h heavy rain event observed in a subsequent experiment.

By analyzing the PSVD matrix, we can derive useful physical properties of the rainfall environment. The particle density (PD), defined as the total number of drops per volume of rainfall space [m^−3^], and the drop size distribution (DSD), which represents the number of drops per unit diameter per volume in the size class $D_{i}$ [m^−3^mm^−1^], are calculated using Equations (1) and (2), respectively. The corresponding precipitation rate *R* [mm/h] is determined using Equation (3):(1)$$PD=\Sigma_{i}PD\left(D_{i}\right)=\frac{1}{At}\sum_{i,j}\frac{n_{i,j}}{V_{j}},$$
(2)$$DSD\left(D_{i}\right)=\frac{1}{At∆D_{i}}\sum_{j}\frac{n_{i,j}}{V_{j}},$$
(3)$$R=\Sigma_{i}R\left(D_{i}\right)=\frac{\pi }{10^{5}At}\sum_{i,j}n_{i,j}{D_{i}}^{3},$$
where *A* is the sampling area [m^2^], *t* is the sampling time [s], $n_{i,j}$ is the number of raindrops as an element (*I*, *j*) of the PSVD matrix, $V_{j}$ is the drop velocity class of the matrix [m/s], and $\Delta D_{i}$ is the bin width of the drop size class $D_{i}$ [mm].

The graph in Figure 2b shows the *DSD* derived from the PSVD matrix. Here, the horizontal axis represents the raindrop size *D* [mm], and the vertical axis represents the number of raindrops per unit volume [m^−3^mm^−1^], for particle sizes ranging from $D_{i}$ to $D_{i}+\Delta D_{i}$, plotted on a logarithmic scale.

The size distribution of raindrops in natural rainfall is known to follow a specific pattern; Marshall and Palmer [10] demonstrated in 1948 that the *DSD* can be effectively parameterized by an exponential function,
(4)$$DSD\left(D\right)=N_{0}e^{-\Lambda D},$$
where $\Lambda =4.1R^{-2.1}$ [mm^−1^], $N_{0}=8000$ [m^−3^mm^−1^], and *R* is the precipitation rate [mm/h].

The Marshall–Palmer distribution tends to overestimate the number of small raindrops. In addition, as raindrops fall from clouds to the ground, the number of small raindrops decreases due to coalescence or adsorption into larger raindrops, as well as evaporation [11]. Therefore, a more general formula involving the gamma function has been proposed [12].

The dotted line in the graph in Figure 2b represents the Marshall–Palmer distribution for R=300. In this observation, the size distribution closely matches the Marshall–Palmer distribution for particle sizes greater than 1 mm, with sizes greater than 6 mm considered outliers.

#### 2.1.3. Visibility of Rainfall

Visibility is the maximum distance at which an object can be clearly seen with the naked eye, and it is an important meteorological observation for aircraft and ship operations. The mechanical measurement of visibility is defined by the World Meteorological Organization (WMO) as the Meteorological Optical Range (MOR), which is “the length of path in the atmosphere required to reduce the luminous flux in a collimated beam from an incandescent lamp, at a color temperature of 2700 K, to 5% of its original value” [13]. MOR [m] can be calculated according to Koschmieder’s law using
(5)$$MOR=d\times \frac{\ln\left(c_{0}\right)}{\ln\left(T_{d}\right)},$$
where $T_{d}$ is the spatial transmittance observed at a distance *d* [m] from the light source, and $c_{0}$ is a constant denoting the threshold of light attenuation. Empirically, $c_{0}$ is 0.02 (2%) in most cases but is set to 0.05 (5%) for rainfall, which is the same as the MOR definition [14]. Throughout this paper, the term “visibility” will refer to the perceptual quality of a space, while “MOR” will refer to the precise physical measurement of visibility.

MOR is commonly used to evaluate sensor performance in fog and similar environments. However, its suitability for technical evaluation of sensor performance is questionable because MOR is tailored to the characteristics of the human eye, as evidenced by the use of different $c_{0}$ values for rainfall and other conditions. Therefore, in our previous work, we demonstrated the advantages of using spatial transmittance over MOR for such evaluations [3].

The spatial transmittance of a rainfall environment can be formulated using Lambert-Beer’s law as
(6)$$T_{d}=e^{-\gamma d}\;,$$
where *γ* [m^−1^] is the absorption coefficient of the rainfall environment, which quantifies the rate at which light is absorbed per unit length in space. This coefficient can be approximated from the PSVD matrix using
(7)$$\gamma =\Sigma_{i}\gamma \left(D_{i}\right)={2\pi \Sigma }_{i}PD\left(D_{i}\right){\left(\frac{D_{i}}{2}\right)}^{2}=\frac{2\pi }{At}\sum_{i,j}\frac{n_{i,j}}{V_{j}}{\left(\frac{D_{i}}{2}\right)}^{2},\;$$
which indicates that the transmittance in a rainfall environment is influenced by the total cross-sectional area of the raindrops rather than the precipitation rate [15]. Observations of actual weather conditions show that drizzle, despite its low precipitation rate, causes less visibility [16]. Conversely, as discussed below, even very high precipitation rates may not significantly reduce spatial transmittance.

In this paper, we demonstrate with experimental results that this formulation is practical. We then elucidate the correlation between raindrop size distribution and sensor detection performance and propose a new metric for sensor evaluation to replace precipitation rate.

In the discussion of visibility in this paper, it is not necessary to consider the wavelength of light. This is because sensor devices operating on optical principles primarily use wavelengths within the visible to near-infrared spectrum, and the spatial transmittance of rainfall does not depend on wavelength across the spectrum [17]. It should be noted, however, that the wavelength should be considered for sensor devices that use light beams beyond the infrared spectrum or in contexts involving aerosols other than precipitation, where the transmittance does exhibit a wavelength dependence.

### 2.2. Standardization

This section reviews international standards related to sensor-performance evaluation and test methods under rain conditions.

The IEC 61496 series is an important international standard for safety sensors, covering technologies such as light curtains, LiDARs, TOF cameras, monocular and Stereo-Vision-based cameras, and radars [2]. This group of standards is primarily intended for indoor factory environments and does not address outdoor conditions, including rain.

For outdoor safety sensors, the IEC 62998 series specifies design requirements. And IEC TS 62998-1 refers to the IEC 60721 series as a quantitative reference for outdoor environments [18].

The IEC 60721 series categorizes the severity of environmental conditions to which electrical and electronic products are exposed, including an extreme rainfall intensity of 6 mm/min (360 mm/h) for short periods of time [19]. However, it lacks details on the physical characteristics of the rain, such as raindrop size distribution or visibility, because it focuses more on product design for environmental resistances than on sensor detection capabilities.

Efforts to standardize ground service robots, including transport and other types, continue under ISO 13482 and its revision projects [20]. While the sensing requirements for these robots include collision avoidance and road surface detection, the standards do not detail specific environmental conditions.

The growing demand for drones has spurred active international standardization, with various guidelines and test methods being developed. For example, ISO 5110 defines test methods for drone flight stability in wind and rain conditions [21]. However, standardization for sensor performance in rain conditions has not been initiated.

In summary, current international standardization efforts are primarily focused on hardware durability in a rainfall environment rather than the impact of rainfall on sensor detection performance. It is anticipated that future regulations will require a focus on the latter. This paper contributes to filling this gap by focusing on particle size distribution and spatial transmittance.

### 2.3. Computer Vision and UAV/UGV

In the fields of computer vision and robotics, a variety of sensing and image-processing techniques have been developed for use in rainy conditions.

Garg and Nayar proposed an efficient algorithm to mitigate the visual effects of rain in both computer vision and computer graphics, based on a detailed analysis [22].

Charette et al. developed a “smart headlight” system. It uses a camera to determine the location of precipitation, such as raindrops and snowflakes, in front of a vehicle. The system then segments the headlight beam with a beam splitter to bypass these particles. This method reduces light scattering and reflection from the particles, thereby improving driver visibility [23].

Lee et al. use sensor fusion techniques with cameras and LiDARs to achieve road detection for autonomous vehicles in various weather conditions, including rain [24]. Murase’s research team has presented several driver assistance technologies. Using cameras and multiple sensors, these technologies accurately detect pedestrians, obstacles, driver status, and weather conditions in rainy and foggy environments [25].

Xi et al. propose a collaborative learning network, CoDerainNet, to improve the accuracy of object detection by drones in rainy weather. The network effectively reduces image noise caused by rain and improves object-detection accuracy and computational cost [26]. Li and Wu propose a framework for extracting highly accurate vehicle motion data from video captured by UAVs. They improve YOLOv5, a typical deep learning algorithm, to improve the accuracy of vehicle detection under various weather conditions, including rainfall [27].

These studies lay the groundwork for both theoretical and practical approaches to improving sensor performance in rainy conditions, which are critical for the deployment of automated machines in outdoor environments.

In addition to these advanced core sensing technologies, evaluation methodologies are also essential to verify the functionality of safety sensor products as intended. This paper addresses this issue by proposing criteria for evaluating sensor performance in rainfall environments.

### 2.4. Our Previous Work

Our research group has actively pursued the evaluation of human detection performance of safety sensors for service robots, leading to the development of a light interference test method under direct sunlight [28] and a simulated snowfall chamber that mimics the visibility reduction due to snow in a room temperature environment [29].

Regarding fog, our experiments with different sensor technologies have revealed significant differences in sensor behavior in foggy conditions. We have demonstrated that fog can drastically affect the detection capabilities of optical sensors to the extent that even minimal fog that is barely perceptible to the human eye can severely degrade the detection capabilities of some sensors [30]. These findings led us to question the suitability of the commonly used visibility metric, MOR, for evaluating sensor performance. As an alternative, we proposed MOT, Minimum Object-detectable Transmittance, a novel metric based on spatial transmittance [3].

In this paper, we extend our focus to include rain and drizzle, which are critical weather phenomena affecting sensor performance along with fog and snowfall. Based on experiments at a large rainfall test facility, we will show that sensor evaluation based on precipitation rates can be misleading in some cases, and we discuss more appropriate methods of performance evaluation.

## 3. Sensor Evaluation in Rain/Drizzle Environments

### 3.1. Precipitation Rate vs. Spatial Transmittance

As mentioned above, precipitation rate is commonly used as an indicator of sensor performance in rain conditions. It is widely recognized as intuitive weather information for rainfall intensity, which is important to sensor users, including designers and integrators working on outdoor drone and robotic applications. Therefore, the specifications for standards-compliant outdoor safety sensors should always include operating limits in precipitation conditions, such as “This sensor product is designed for use in precipitation rates up to 80 mm/h”.

However, relying solely on precipitation rate to measure sensor performance is fundamentally flawed. The precipitation rate measures the amount of water that falls to the ground and accumulates, losing information about the physical properties of the precipitation space between the sensor and the target. In fact, as previously discussed, a high precipitation rate does not always correlate with poor visibility.

Sensor performance should be evaluated based on the physical characteristics of the space, i.e., the “visibility”. To quantitatively evaluate the visibility, this paper adopts the spatial transmittance $T_{d}$ at distance *d* [m] intended for sensor use, as introduced in a previous study [3]. To evaluate the visibility of the rainfall space itself without relying on a specific sensor, we use $T_{5}$, which uses 5 m as a representative distance. For added compatibility, the MOR calculated by Equation (5) can also be used in conjunction with spatial transmittance.

### 3.2. Object-Detection Rate in Rainfall Environments

This section presents the methodology used to measure the detection capability of the sensor under rain conditions, focusing on the object-detection rate. A test piece is strategically placed at $d_{S}$ meters within the field of view of the sensor *S* in a controlled precipitation condition, *PC*, and observations and detections are executed repeatedly over *f* frames. The detection rate, *DR*, is then simply defined as follows:(8)$$DR\left(S,d_{s},PC\right)=\frac{f_{det}}{f}=\frac{f_{det}}{f_{det}+f_{fail}},$$
where $f_{det}$ is the number of frames out of all *f* frames in which the test piece is successfully detected, while $f_{fail}$ is the number of frames in which detection failed.

We then investigated how the physical properties of rain, such as precipitation rate and particle density, affect the performance of the sensor. This is carried out by comparing the detection rate $DR\left(S,d_{s},PC0\right)$ of the sensor *S* in a clear environment with no precipitation to the detection rate $DR\left(S,d_{s},PCx\right)$ under varying precipitation conditions *PCx*, which represent different combinations of precipitation rate and particle density.

### 3.3. Simulated Precipitation Conditions

The experiments described in this paper were conducted at the Large-scale Rainfall Simulator, a test facility of the National Research Institute for Earth Science and Disaster Resilience (NIED) [31]. A photograph of the facility is shown in Figure 3a. This facility is one of the world’s largest and most powerful water spray facilities capable of reproducing nature-like rainfall and has the following specifications:Nozzles with four different spray diameters installed 16 m above the ground (2176 nozzles in total);Capable of spraying up to approximately 3000 m^2^ (44 m × 72 m);Maximum precipitation rate of approximately 300 mm/h;Raindrops up to approximately 6 mm in diameter.

Using a nozzle with a smaller spray diameter and increasing the water pressure (i.e., increasing the precipitation rate) increases the distribution of the number of small diameter raindrops. By changing the combination of nozzle and water pressure, it is possible to reproduce rainfall with different precipitation rates and drop size distributions.

In this experiment, three of the four nozzle types were used to reproduce four different precipitation conditions, which were expected to have very different combinations of precipitation and transmittance (particle density) from each other, in order to facilitate verification of the claims made in this paper. Measurements were made under the following five precipitation conditions, including the clear condition:*PC*0: Clear condition with no rainfall.*PC*1: Rainfall at a pre-set precipitation rate of 45 mm/h using Nozzle 1, which has the smallest spray diameter. Precipitation consisting mainly of drizzle-sized particles is reproduced.*PC*2: Rainfall at 80 mm/h using Nozzle 2, which has a medium-diameter spray nozzle. Reproduces heavy rainfall.*PC*3: Rainfall at 200 mm/h with Nozzle 2. Reproduces a cloudburst with many drizzle-sized particles.*PC*4: Rainfall at 300 mm/h using Nozzle 4, which has the largest spray diameter. Reproduces the largest cloudburst consisting mainly of large raindrops.

The particle size and velocity distribution, PSVD, of the simulated rainfall was measured using an OTT Parsivel2 laser precipitation disdrometer (OTT HydroMet, Kempten, Germany), an instrument widely used in both practical and research applications in meteorology.

In this paper, we describe two types of precipitation rates for rainfall simulated at the facility:Values pre-set by the facility;Values calculated from the observed PSVD matrix using Equation (3).

Due to the inherent non-uniformity of the simulated rainfall, these two values often do not coincide and can sometimes differ significantly.

### 3.4. Test Piece and Experimental Setup

In this experiment, the test piece for sensor detection was a matte black acrylic plate with dimensions of H: 400 mm × W: 200 mm. The width of 200 mm is indicative of the minimum human torso size considered in the design of safety sensors, as specified in IEC 61496-3 [32].

The diffuse surface reflectance of the test piece under dry conditions was approximately 5% across the visible to near-infrared spectrum. During the rainfall experiments, the test piece was exposed to rain, and water droplets were deposited on its surface. To ensure consistent surface conditions across all experiments, including those under *PC*0 without rain, the test piece surface was intentionally maintained with water droplets, as shown in the photograph in Figure 4a.

This black test piece with water droplets presents a challenging detection target for optical sensors. In fact, a preliminary evaluation revealed that some sensor products could not detect the test piece even under the clear condition *PC*0 (these products were subsequently excluded from the main experiment).

The positioning of the test piece and the sensor is shown in Figure 4b. To prevent water droplets from affecting the optical window of the sensor as described in Section 1.1 and to prevent electrical damage from water, the sensor was placed under a tent to ensure a distance of 500 mm between the optical window and the rainfall. Consequently, the portion of the distance *d* between the test piece and the sensor’s optical window that was exposed to the rain was (d−500) mm. The test piece was oriented perpendicular to both the ground and the optical axis of the sensor. The height of the bottom edge of the test piece and the optical axis of the sensor was set to 1000 mm above the ground reference plane.

The selection of the distance $d_{S}$ from the sensor’s optical window to the test piece required careful consideration due to the varying basic performance of the sensors used in this experiment. We determined *d* to be the maximum distance at which the sensor could reliably detect the test piece in a clear environment. This was operationalized as finding a position where the detection rate $DR(S,d_{S},PC0)$ for the sensor *S* in the clear environment *PC*0 does not reach 100% but remains as high as possible. This implies finding a distance $d_{S}$ that satisfies
(9)$$\sup\left\{DR\left(S,d_{S},PC0\right)|d_{S}\in A\right\}<1.0,$$
where *A* represents the set of distances at which the sensor *S* can detect the test object in the clear environment *PC*0.

With this setup, the impact of rain on the sensor can be sensitively reflected in the detection rate. That is, if the sensor *S* can almost detect the target at *PC*0, i.e., $DR\left(S,d_{S},PC0\right)\approx 1.0$, and the detection rate decreases significantly under precipitation conditions *PC*x, i.e., $0.0\leq DR\left(S,d_{S},PCx\right)\ll 1.0$, then the effect of rainfall *PC*x on the sensor can be estimated from these differences. Conversely, if the *DR* is hardly reduced by *PC*x, then the sensor is hardly affected by rainfall.

### 3.5. Sensor Devices

In this experiment, we tested the following four depth cameras, each based on one of the three sensing technologies described in Section 1.1, to investigate how the impact of rainfall varies with the sensing technology:*S*1: Stereo Vision (SV);*S*2: Structured Light (SL);*S*3: Time of Flight (TOF);*S*4: Time of Flight (TOF).

It should be noted that all sensors are general-purpose products, not designed for safety applications, and were not originally intended for use in a rainy environment. Therefore, since none of the sensors are considered to be technically prepared for rainfall, we expect to be able to directly observe the effects of rainfall on the performance of the sensor devices.

The detection rates described in Section 3.2 were calculated for the depth images produced by these sensors. In order to measure the detection rate under the same conditions for all sensors, we used software implementing our original object-detection algorithm instead of the proprietary software bundled with the sensors. The software is available online under an open-source license as a common platform for object detection [33].

In this experiment, the depth image frames continuously output by the sensor were processed using this common software to identify candidate object regions. Object detection was considered successful if the area of the candidate region reached 400 pixels or more in the image coordinate system. Such frames were counted as successful detection frames $f_{det}$ in Equation (8). However, for sensor *S*1, which has a wide viewing angle, the threshold was lowered to 100 pixels due to the relatively small object area compared to the depth image size. Areas detected away from the position of the test piece, such as the ground and the tripod legs supporting the test piece, were excluded using the masking function of the software. For more details, refer to [33].

## 4. Experimental Results

### 4.1. Physical Properties of Precipitation

Table 1 shows the measurement results for the following physical properties of rainfall reproduced under the four predefined precipitation conditions:Precipitation rate, *R* (pre-set and measured);Particle density, *PD* (calculated by Equation (1));Absorption coefficient *γ*, the corresponding 5 m transmittance $T_{5}$ and MOR (Equations (5)–(7)).

Figure 5 shows plots of the drop size distribution (DSD) for each of the precipitation conditions and the distribution of the absorption coefficient *γ*, as listed in Table 1, for each raindrop particle size class.

These data show the following:
The 45 mm/h rainfall in *PC*1 was characterized by mainly drizzle mixed with rain, with the majority of drops being 1 mm or less in particle size. Despite having the lowest precipitation rate among the four types, its particle density, *PD*, was about 65,000, which was not significantly different from *PC*3 at 200 mm/h. The absorption coefficient was also the second highest among the four types, with a 5 m transmittance, $T_{5}$, of 79.8% (corresponding MOR: 66.6 m), making the visibility in this condition the second worst after *PC*3.The 80 mm/h rainfall in *PC*2 is characterized as typical heavy rainfall. However, its *PD* was the lowest among the four types, being less than half of that observed in *PC*1. The absorption coefficients were significantly lower than those in *PC*1 for all particle sizes. In addition, the $T_{5}$ was 92.4% (MOR: 189.6 m), making it almost indistinguishable from conditions that are clear to the human eye at close range.The 200 mm/h rainfall in *PC*3 had a distribution of raindrops of 1 mm or less in particle size similar to that of *PC*1, but it reached a cloudburst level due to the significantly higher number of 1 to 3 mm raindrops. The $T_{5}$ was 71.2% (MOR: 44.2 m), which was the worst visibility among the four types, although the difference from *PC*1 was not substantial. This suggests that the contribution of light absorption by raindrops larger than 1 mm in particle size is relatively small.The 300 mm/h rainfall in *PC*4 had cloudburst-class conditions, with a noteworthy presence of raindrops of 2 to 5 mm in particle size. However, its *PD* was comparable to that of *PC*2, and the $T_{5}$ of 86.6% (MOR: 104.3 m) was not significantly different. As shown in the absorption coefficient plot, the coefficients for small raindrops of 1 mm or less in particle size were markedly lower than those observed for *PC*1 and *PC*3, while larger raindrops of 2 mm or more in diameter had a non-negligible but limited impact. Consequently, the impact of this rainfall on visibility was relatively moderate.

In summary, these findings indicate that high precipitation rates do not necessarily result in reduced visibility within the precipitation space. The graphs in Figure 5 clearly show that raindrops with a particle size of 1 mm or less are the primary contributors to reduced visibility in the rainy environment. This is consistent with the observations of the low visibility from natural drizzle discussed in Section 2.1. This also suggests that the selection of the precipitation conditions described in Section 3.3 was appropriate. It also reinforces the argument made in this paper that the precipitation rate alone is not an appropriate metric for assessing sensor performance.

### 4.2. Evaluation of Sensor Detection Performance

#### 4.2.1. Evaluation Procedures

This section details the results of the object-detection tests conducted with the sensors and the precipitation conditions mentioned above.

First, detection was performed under *PC*0, a clear condition with no precipitation, to determine the distance *d* as described in Section 3.2. Accurately determining the distance *d* that satisfies Formula (9) proved to be difficult. Therefore, we incrementally adjusted the position of the test piece during detection. This approach led to an ad hoc determination of the maximum distance at which the detection rate dropped below 100%. The distances thus determined for each sensor and the corresponding detection rate DR(d,PC0) are listed in Table 2.

For sensor *S*4, we could not experimentally find a distance *d* at which a detection rate $DR\left(S4,d,PC0\right)$ close to 100% could be obtained stably and at which the detection rate would decrease under rain conditions. This is because *S*4 was originally designed for indoor short-range sensing applications, and its operation was very unstable in this experiment. Therefore, we substituted the detection rate *DR* of 67.37% obtained at the distance *d* of 3596 mm. Since this does not strictly satisfy Formula (9), the results for sensor *S*4 could be considered as supplementary information.

Figure 6 shows examples of depth images and RGB images from frames where each sensor achieved successful detections in the clear condition, *PC*0. In the depth images, pixels closer to the sensor appear in brighter gray or red, while pixels without distance information are shown in black. The red areas indicate object regions identified by the object-detection algorithm discussed earlier, and such frames are categorized as successful detection frames $f_{det}$.

Then, for each sensor *S*, the test piece was positioned at the specified distance $d_{S}$, and detection tests of more than 500 frames were executed repeatedly both in the clear condition *PC*0 and in precipitation conditions *PC*1 to *PC*4.

#### 4.2.2. Detection Rates under the Precipitation Conditions

Figure 7 shows a graph summarizing the detection rates achieved in the experiment. Figure 8 and Figure 9 show examples of successful and unsuccessful detections by each of the sensors in precipitation conditions *PC*1 (45 mm/h) and *PC*4 (300 mm/h), respectively.

As shown in Figure 7, the *PC*1 (45 mm/h) rainfall, with the lowest precipitation rate of the four types, had the most significant impact on the sensors, causing all sensors to fail to detect the test piece. Conversely, the *PC*4 (300 mm/h) rainfall had a significantly lower impact than *PC*1, despite being one of the heaviest natural rainfalls. The RGB images in Figure 8 and Figure 9 clearly illustrate that *PC*1 has significantly poorer visibility.

Similarly, the *PC*3 rainfall (200 mm/h) resulted in near-zero detection rates for all sensors, demonstrating that its effect on sensor performance is comparable to that of *PC*1. However, since the precipitation rate of *PC*3 is much higher than that of *PC*1, this result further suggests that precipitation rate has minimal effect on sensor performance.

These results clearly demonstrate that rainfall intensity alone is not a reliable metric for sensor detection capability. Regardless of the sensing technology used, it is primarily the spatial transmittance, determined by the particle density *PD* of small raindrops, that affects a sensor’s detection performance.

Examples of detection failures shown in Figure 8 and Figure 9 include the following cases: For sensors *S*1 and *S*3, when the area representing the test piece falls below the detection threshold;For sensor *S*2, when it is unable to acquire distance data due to rain;For sensor *S*4, when the test piece is obscured by noise and cannot be detected as a connected region.

#### 4.2.3. Sensing Technologies under the Precipitation Conditions

This section analyzes whether there are differences in the effect of rainfall on different sensing techniques.

For precipitation conditions *PC*2 (80 mm/h) and *PC*4 (300 mm/h), which are characterized by a lower number of drizzle particles, a significant difference in the detection performance of the sensors was observed.

Interestingly, the detection rate of the TOF sensor *S*3 did not decrease significantly under the *PC*2 and *PC*4 precipitation conditions, although previous reports indicated that TOF-based sensors tend to be highly affected by scattering from particles in space [30,34]. As shown in Figure 5, *PC*1 (45 mm/h) and *PC*3 (200 mm/h) are significantly affected by scattering effects due to small raindrops of 1 mm or less in particle size, while *PC*2 and *PC*4, which contain significantly smaller numbers of particles, are less affected.

The marked difference in detection rates between the *S*2 and *S*4 sensors, both of which use the TOF method, could be attributed to the fact that the *S*4 sensor is designed for short-range indoor use, which makes it difficult to detect low-reflectivity test pieces even under clear *PC*0 conditions, thus making it more susceptible to rainfall.

The detection rate was higher for *PC*4 compared to *PC*2 in the case of *S*2 using SL and *S*4 using TOF, even though the spatial transmittance of *PC*4 was slightly lower than that of *PC*2, but not by a significant margin. This does not easily explain why *PC*4 would have a higher detection rate. The fact that both sensors are intended for indoor applications may contribute to the considerable variability in performance at their limits.

## 5. Discussion

This section presents an analysis showing that small raindrops cause a more significant problem for sensors and discusses a metric for evaluating sensor performance in rainfall environments.

### 5.1. Raindrop Size and Spatial Transmittance

Figure 10 graphically illustrates the precipitation rate required to achieve the same spatial transmittance as the *PC*1 (45 mm/h) precipitation condition described in Section 3.3, but in a hypothetical scenario where all raindrops are of uniform particle size. The number of raindrops $PD(D_{i})$ required to achieve the same 5 m transmittance $T_{5}=0.798$8 (MOR: 66.6 m) as *PC*1 by rainfall consisting only of raindrops of particle size $D_{i}$ [mm] was determined using Equations (6) and (7). And the fall velocity $V_{i}$ of raindrops of particle size $D_{i}$ was estimated by the Foote and Du Toit polynomial [35]. Then, the precipitation rate $R_{i}$ calculated by Equations (2) and (3) was plotted.

According to these calculations, a precipitation rate consisting of drizzle-sized raindrops with a particle size of 0.4 mm, which has a spatial transmittance equivalent to *PC*1, is approximately 35 mm/h. This is consistent with the experimental results, because the 0.4 mm particle size corresponds to the most frequent raindrop size of raindrops in *PC*1, as shown in Figure 5.

However, it would require a cloudburst-level rainfall of about 216 mm/h, consisting of only 1 mm raindrops, to match the transmittance of *PC*1. Obviously, such a precipitation rate cannot be produced by 1 mm raindrops alone. In the case of 3 mm raindrops, it would require a precipitation rate of approximately 1308 mm/h, which is unrealistic for Earth.

### 5.2. Redefining Sensor Evaluation Metrics for Rainfall

From the above analysis, it is clear that using precipitation rate as a metric to evaluate sensor performance is not appropriate. Instead, the spatial transmittance $T_{d}$ at the measurement distance *d* where the sensor is intended to be used, or the density of small particles PD(D|D≤1.0), should serve as the metric for sensor evaluation. More specifically, since the density of larger particles, PD(D|D>1.0), is negligibly small, the total particle density *PD* can be used as an indicator, where $PD\left(D|D\leq 1.0\right)≅PD$.

To provide information to sensor users, such as safety sensor system integrators, a description based on the intended maximum operating distance and its spatial transmittance, as suggested in our previous paper [3], should be used. For example, the user manual should state: “This sensor is designed for use in rainfall with a spatial transmittance greater than 80% at a sensing distance of 5 m”. Alternatives might include “… with a raindrop density *PD* of 50,000 or less” or “… with an MOR of 65 m or more”.

If there is a need to use precipitation rate as a metric for compatibility or ease of general understanding, it is important to distinguish between rainfall and drizzle. For example, “… for use in rainfall of less than 80 mm/h or in drizzle of less than 0.5 mm/h”.

Relying on precipitation rate alone should be avoided, as it can lead to inaccuracies under certain environmental conditions.

## 6. Conclusions

This study investigated the object-detection performance of non-contact safety sensors in rainfall environments. Our experimental results highlighted that higher precipitation rates do not necessarily correlate with reduced visibility and that the presence of numerous small raindrops has a greater impact on sensor detection performance than the sheer volume of precipitation. These results challenged the conventional use of precipitation rate as a metric for evaluating sensor performance, providing an understanding of its inadequacy and potential for misleading conclusions in certain scenarios. We recommended the adoption of spatial transmittance or raindrop density as more reliable indicators for assessing sensor effectiveness in precipitation conditions.

In this experimental design, we selected four types of optimal precipitation conditions given our limited resources. Ideally, experiments should include a wider variety of precipitation conditions and a greater number of sensor devices. Additionally, validating the approach through field tests is crucial. This will further reinforce the proposed methodology. These are challenges to be addressed in future research.

Integrating the findings from this research with our previous work [3,28,29,30], we conclude that spatial transmittance is a universally applicable metric for evaluating sensor performance across a spectrum of environmental conditions, including rain, drizzle, and fog. This conclusion leads us to propose that spatial transmittance may be the only metric worth considering for evaluating sensor object-detection performance, regardless of the specific weather phenomena. In addition, the applicability of this metric could potentially be extended to various aerosols and environmental particles such as blowing snow and sand dust. However, the influence of spatial transmittance on sensing capabilities can vary significantly depending on the particle types present, suggesting that the uniform application of this metric requires careful consideration and further validation. Addressing these nuanced challenges and testing the proposed hypotheses will be essential for future research in this field.

## Figures and Tables

**Figure 1 sensors-24-02713-f001:**
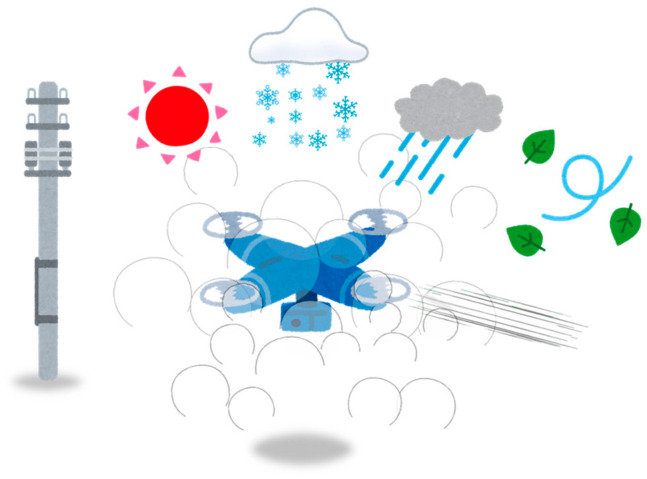
UAV in adverse weather conditions.

**Figure 2 sensors-24-02713-f002:**
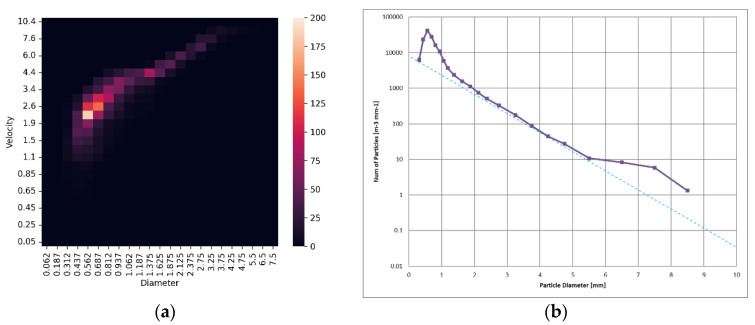
PSVD Matrix and DSD measurements from the NIED Large-scale Rainfall Simulator on 31 October 2023. (**a**) PSVD matrix depicting 300 mm/h cloudburst (raindrop counts per 10 s visualized as a heat map); (**b**) DSD derived from the PSVD matrix, overlaying the Marshall Palmer distribution with *R* = 300.

**Figure 3 sensors-24-02713-f003:**
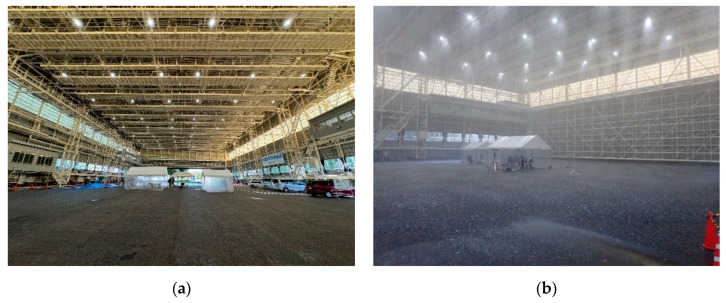
Photographs of the Large-scale Rainfall Simulator, a test facility of NIED. (**a**) Inside the facility; (**b**) rainfall simulated for precipitation condition *PC*4 (300 mm/h).

**Figure 4 sensors-24-02713-f004:**
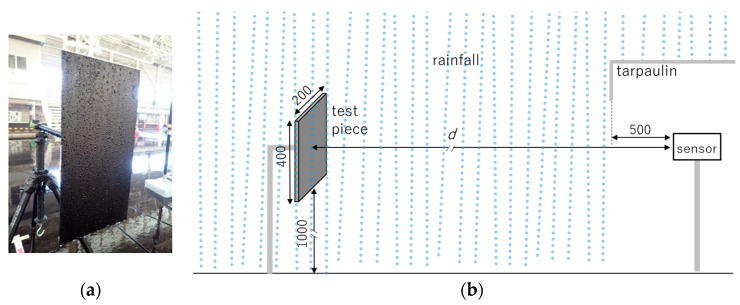
Experimental setup. (**a**) Test piece with raindrops on surface; (**b**) positioning of the test piece and a sensor.

**Figure 5 sensors-24-02713-f005:**
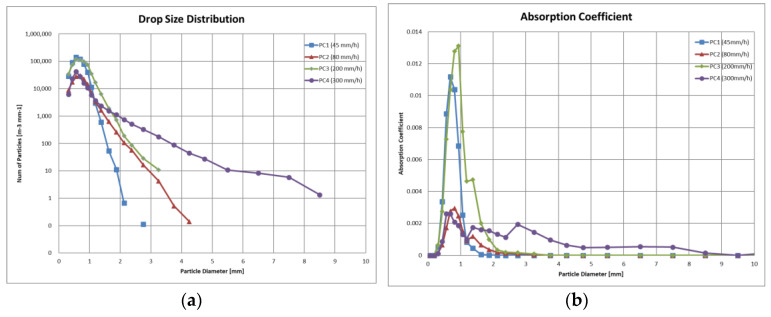
Measurement results of the physical properties for each raindrop particle size class. (**a**) Drop size distribution, *DSD*. (**b**) Absorption coefficient, *γ*.

**Figure 6 sensors-24-02713-f006:**
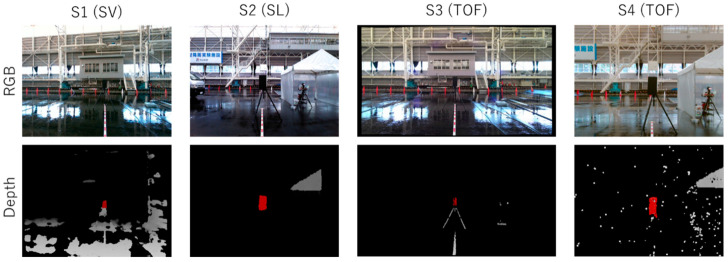
Examples of output images of sensors *S*1–*S*4 in the clear condition, *PC*0. Top: RGB images. Bottom: depth images.

**Figure 7 sensors-24-02713-f007:**
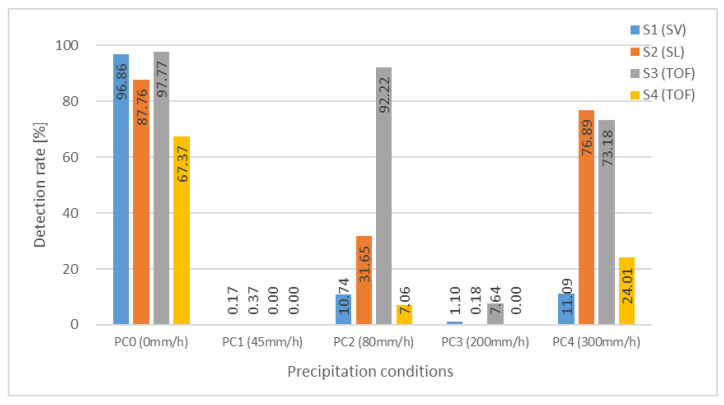
Detection rates for each of the sensors under the precipitation conditions.

**Figure 8 sensors-24-02713-f008:**
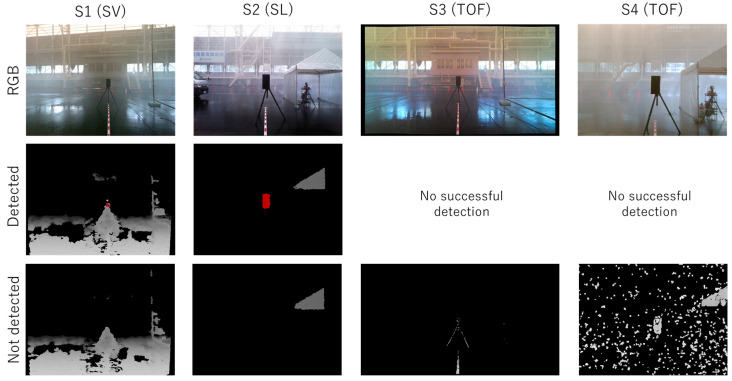
Example output images of the sensors *S*1–*S*4 in precipitation condition *PC*1 (45 mm/h). Top: RGB images. Middle: depth images showing cases where the test piece was detected. Bottom: depth images where the test piece was not detected.

**Figure 9 sensors-24-02713-f009:**
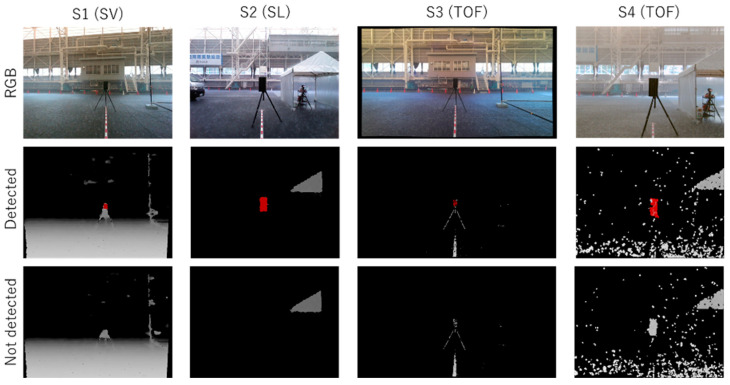
Example output images of sensors *S*1–*S*4 in precipitation condition *PC*4 (300 mm/h). Top: RGB images. Middle: depth images showing cases where the test piece was detected. Bottom: depth images where the test piece was not detected.

**Figure 10 sensors-24-02713-f010:**
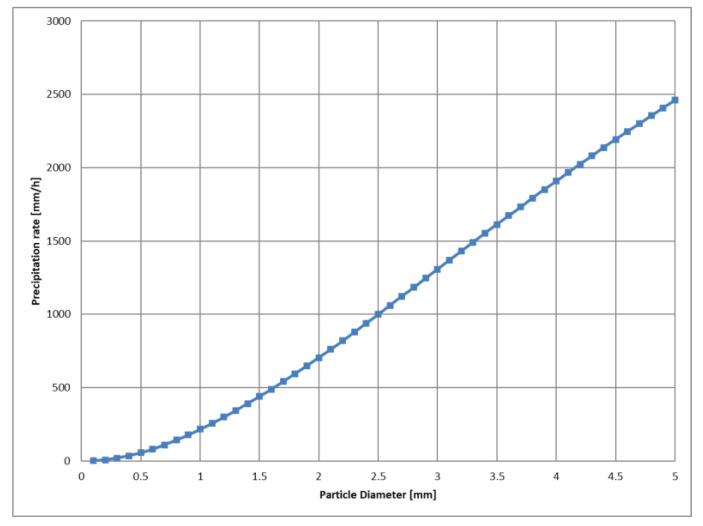
Precipitation rates required to achieve a 5 m spatial transmittance $T_{5}$ of 79.8%, assuming uniform raindrop particle sizes.

**Table 1 sensors-24-02713-t001:** Measurement results of the physical properties of rainfall for each of the predefined precipitation conditions.

Precipitation Condition	Precipitation Rate, *R* [mm/h], Pre-Set/Measured	Particle Density, *PD* [m^−3^]	Absorption Coefficient, *γ* [m^−1^]	Spatial Transmittance (5 m), *T*_5_ [%]	MOR [m]
*PC*1	45/46.2	64,659.2	0.0451	79.8	66.6
*PC*2	80/59.8	17,043.2	0.0158	92.4	189.6
*PC*3	200/118.4	72,718.7	0.0678	71.2	44.2
*PC*4	300/334.1	18,949.4	0.0288	86.6	104.3

**Table 2 sensors-24-02713-t002:** Measurement distances, *d*, determined for the sensors.

Sensor	Sensing Technology	*d* [mm]	*DR* (*d*, *PC*0) [%]
*S*1	SV	4210	96.86
*S*2	SL	3231	87.76
*S*3	TOF	4101	97.77
*S*4	TOF	3596	67.37 *

* For sensor *S*4, *DR* (*PC*0) is not close to 100%, which does not strictly satisfy Formula (9).

## Data Availability

Data are contained within the article. Software is online available.

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
