# Peer review of "Impact of Rainfall on the Detection Performance of Non-Contact Safety Sensors for UAVs/UGVs"

_sensors, 2024, doi:10.3390/s24092713_

Round 1

Reviewer 1 Report

Comments and Suggestions for Authors

The paper is very interesting and the research topic is impacting the community. 

However, the paper needs some revisions that the Authors have to do.

1. The introduction is not reporting in clear the aims of the paper. There is some confusions about the research topic, experimental results, tests, sensors, ...etc. I would expect that the authors writes in clear the main contributions and the aims of their research.

2. Since the manuscript covers 18 pages, it would be beneficial to have at the end of the introduction section a paragraph describing the main sections of the manuscript.

3. Lines 295 to 299 at the end of Section 2, the authors do some statements that have to be harmonized with those described in Section 1.

4. Does the Figure 9 and its caption describes the correct results?

Best Regards,

The Reviewer 

Reviewer 2 Report

Comments and Suggestions for Authors

The authors design and run an experimental procedure to measure the robustness of 4 non-contact sensors (S1: Stereo Vision; S2: Structured Light; S3: Time-of-Flight; and, S4: Time-of-flight) in terms of object detection performance. Robustness is measured in terms of an object detection rate under the influence of rainfall and drizzle. The main argument put forward by the authors is that the spatial transmittance and particle density measures are better metrics than the precipitation rate. They indicate that a high precipitation rate does not necessarily result in reduced visibility. In Table 1, MOR measurements are reported to justify that the precipitation rate cannot be used alone to provide sufficient information about this metric. It is shown that spatial transmittance and particle density are better metrics for this task.

One obvious criticism is that the authors should comment on the fact that only 4 experiments are presented (each corresponding to a different precipitation rate). Are the supplied experiments sufficient to back up the conclusions regarding the suitability of spatial transmittance and particle density? The authors should clarify this aspect in their discussion, or augment their experimental section with more experiments.

In the particular experiment (see Figure 7 in Section 4.2.2) regarding the PC0 precipitation condition, sensor S4 should have a quantitatively higher detection rate, particularly higher than 67.37%. The lowest object detection rate over the other 3 sensors is 87.76%, which is by 20% higher than the object detection rate of sensor S4. In order to have a better detection rate, maybe the authors could try to shorten the distance from the sensor to the object.

The related work section could be extended with more similar previous studies.

Comments on the Quality of English Language

- In Table 1, there is space character after the 3rd digit, which is certainly wrong.

- "Regarding on fog" should be replaced by "Regarding fog".

-  Before Equation 6, the sentence "The relationship between transmittance and raindrop size distribution [...]" should be corrected.
